# Development of a neural network model to predict the presence of fentanyl in community drug samples

Lianping Ti[1,2]*, Cameron J. Grant[1], Samuel Tobias[1,3], Dennis K. Hore[4,5], Richard Laing[6], Brandon D. L. Marshall[7]

1 British Columbia Centre on Substance Use, Vancouver, British Columbia, Canada, 2 Department of Medicine, University of British Columbia, Vancouver, British Columbia, Canada, 3 School of Population and Public Health, University of British Columbia, Vancouver, British Columbia, Canada, 4 Department of Chemistry, University of Victoria, Victoria, British Columbia, Canada, 5 Department of Computer Science, University of Victoria, Victoria, British Columbia, Canada, 6 Strategic Research and Science Development, Health Canada Drug Analysis Service, Burnaby, British Columbia, Canada, 7 Department of Epidemiology, Brown University School of Public Health, Providence, Rhode Island, United States of America

☯ These authors contributed equally to this work.
* bccsu-lt@bccsu.ubc.ca

**Data Availability Statement:** All data table files containing wavenumber and corresponding absorbance for each spectrum (i.e., drug sample) and corresponding immunoassay strip results are

## Abstract

### Introduction

Increasingly, Fourier-transform infrared (FTIR) spectroscopy is being used as a harm reduction tool to provide people who use drugs real-time information about the contents of their substances. However, FTIR spectroscopy has been shown to have a high detection limit for fentanyl and interpretation of results by a technician can be subjective. This poses concern, given that some synthetic opioids can produce serious toxicity at sub-detectable levels. The objective of this study was to develop a neural network model to identify fentanyl and related analogues more accurately in drug samples compared to traditional analysis by technicians.

### Methods

Data were drawn from samples analyzed point-of-care using combination FTIR spectroscopy and fentanyl immunoassay strips in British Columbia between August 2018 and January 2021. We developed neural network models to predict the presence of fentanyl based on FTIR data. The final model was validated against the results from immunoassay strips. Prediction performance was assessed using F1 score, accuracy, and area under the receiver-operating characteristic curve (AUROC), and was compared to results obtained from analysis by technicians.

### Results

A total of 12,684 samples were included. The neural network model outperformed results from those analyzed by technicians, with an F1 score of 96.4% and an accuracy of 96.4%, compared to 78.4% and 82.4% with a technician, respectively. The AUROC of the model was 99.0%. Fentanyl positive samples correctly detected by the model but not by the

available from the following public repository on GitHub: https://github.com/DrugCheckingBC/nn-fentanyl-prediction.

**Funding:** The study was supported by the Health Canada Substance Use and Addictions Program (1718-HQ-000024; https://www.canada.ca/en/health-canada/services/substance-use/canadian-drugs-substances-strategy/funding/substance-use-addictions-program.html), Vancouver Foundation (https://www.vancouverfoundation.ca), and the US National Institutes of Health-National Institute on Drug Abuse (R01DA052381; https://nida.nih.gov). The content is solely the responsibility of the authors and does not necessarily represent the official views of these funding agencies. LT is supported by a Michael Smith Health Research British Columbia (MSHRBC; https://healthresearchbc.ca) Scholar Award. The funding organizations had no role in the design, data collection, analysis, or preparation of the manuscript. The co-author (RL) who is affiliated with Health Canada is not a co-investigator of the grants that funded our study.

**Competing interests:** The authors declare that no competing interests exist.

technician were typically those with low fentanyl concentrations (median: 2.3% quantity by weight; quartile 1–3: 0.0%-4.6%).

## Discussion

Neural network models can accurately predict the presence of fentanyl and related analogues using FTIR data, including samples with low fentanyl concentrations. Integrating this tool within drug checking services utilizing FTIR spectroscopy has the potential to improve decision making to reduce the risk of overdose and other negative health outcomes.

## Introduction

Recent decades have seen a rapid shift in the unregulated drug market, with the emergence of novel synthetic opioids (e.g., fentanyl, carfentanil) and their associated harms (e.g., overdose, mortality) [1–3]. In response, drug checking services have been implemented as an important harm reduction intervention for people who use drugs to chemically analyze their substances and receive fact-based information and consultation regarding the compounds detected in their sample [4–6]. Not a new concept, the vast majority of drug checking services have previously focused on drugs used in party and festival settings (e.g., psychedelics, stimulants) [7, 8], so less is known about drug checking in the context of the synthetic opioid-driven overdose epidemic. While limited, recent studies have started to show a positive impact of drug checking, including increased engagement in overdose risk reduction practices following the use of the service in some communities [5, 9–11].

Drug checking can be conducted using a range of technologies, offering both benefits and challenges in terms of accuracy, timing, cost, portability, ability to detect a range of compounds, preparation, and required human resources [4, 6, 12, 13]. For example, gas chromatography/mass spectrometry (GC-MS) is highly sensitive, but is expensive, time-consuming, and requires a trained laboratory technician to operate the machine [6, 14]. In contrast, immunoassay strips are inexpensive, easy to use, portable, and can provide results in less than five minutes, but they are unable to detect a range of compounds and can only produce qualitative results (i.e., presence, absence) [15, 16]. Fourier-transform infrared (FTIR) spectroscopy is being implemented across North America for drug checking given the ease and speed of sample preparation and analysis, portability, and ability to detect a range of compounds [15, 16]. However, recent validation studies have indicated that FTIR spectroscopy can only detect compounds above a certain limit; for fentanyl and some fentanyl analogues, that limit is 3–10% quantity by weight, depending on the setting and the presence of specific cuts and buffs [16, 17]. Given that some analogues (e.g., carfentanil) can cause toxicity at very low concentrations [18, 19], failing to detect these compounds when present is a substantial concern when using FTIR spectroscopy on its own. Additionally, there is some subjectivity in interpreting FTIR spectra by drug checking technicians that could result in differing outcomes. Previous studies have shown that when compared to laboratory reference standards, analysis using FTIR spectroscopy can have false negative rates as high as 27.9% [15]. As a result of this limitation, many settings have adopted an approach that utilizes a combination of both FTIR spectroscopy and immunoassay strips (which have a lower detection limit) in tandem to offset the limitations of each technology when used alone [20–22]. However, this can come with its own set of logistical challenges, including additional costs, sample preparation, and timing.

Machine learning applications in the field of health research have been steadily growing due to advancements in processing power and data sharing [23–25]. In recent years, deep learning methods have gained popularity in predictive modelling for substance use research [26]. For example, recurrent neural networks utilizing electronic health records have been applied to classify people who use opioids as long-term patients, short-term patients, and opioid-dependent patients [27]. Other studies have demonstrated the ability of long short-term memory models, an application of deep learning, to accurately predict new onset and prevalence of opioid use disorder using electronic health records [28].

Despite this promising line of inquiry, little is known about whether there is utility in applying deep learning methods to improve the accuracy of FTIR spectroscopy in detecting fentanyl and its analogues, especially when fentanyl is present at low concentrations. The present study aimed to develop a machine learning model that could predict the presence of fentanyl and related analogues with greater accuracy compared to analysis of spectra by a drug checking technician, with a second aim of the model being able to detect fentanyl at low concentrations. Building tools that can be used to accurately detect fentanyl and its analogues is key to ensuring that point-of-care drug checking technologies are providing people who use drugs with the most accurate information to make decisions about their use.

## Methods

### Study setting

British Columbia (BC), the westernmost province of Canada, has been disproportionately affected by the opioid-driven overdose epidemic [3]. By the time a public health emergency was declared in 2016, a significant number of lives had already been lost to illicit drug toxicity, largely due to the widespread presence of fentanyl and other novel synthetic opioids in the unregulated drug supply [29]. In response, a diverse range of harm reduction interventions, including supervised consumption sites, take-home naloxone programs, and drug treatment have been scaled-up [30–32]. As a result, illicit drug toxicity death rates were decreasing between 2018 and 2019 from 31.2 per 100,000 to 19.4 per 100,000, respectively; however, the emergence of the COVID-19 pandemic and corresponding social distancing and self-isolation measures resulted in a significant rise in death rates from 34.4 per 100,000 in 2020 to 44.1 per 100,000 in 2021 [3, 33].

### Data sources

The Drug Checking BC Database includes de-identified, point-of-care drug checking data collected utilizing Bruker (Billerica, Massachusetts, United States of America) ALPHA FTIR spectrometers and BTNX (Markham, Ontario, Canada) immunoassay strips from over a dozen participating community harm reduction sites in BC. Further details about the drug checking service and the use of these technologies, which have been operating since October 2017, have been described elsewhere in detail [15, 20]. Briefly, individuals in harm reduction sites anonymously provided an approximately 5 mg drug sample to a trained drug checking technician and information about what they believed the sample to contain [34]. The drug sample was then analyzed with FTIR and immunoassay strips, with results being immediately available to the person who provided the sample. Results from FTIR analysis include compounds detected in abundant amounts (typically those 5% by weight and above), including both active ingredients (e.g., fentanyl, heroin, methamphetamine) and inert cutting agents (e.g., mannitol, inositol, lactose) [6, 17, 20]. An illustrative example of what a drug checking technician analyzes (and attempts to determine if fentanyl is present) is displayed in Fig 1.

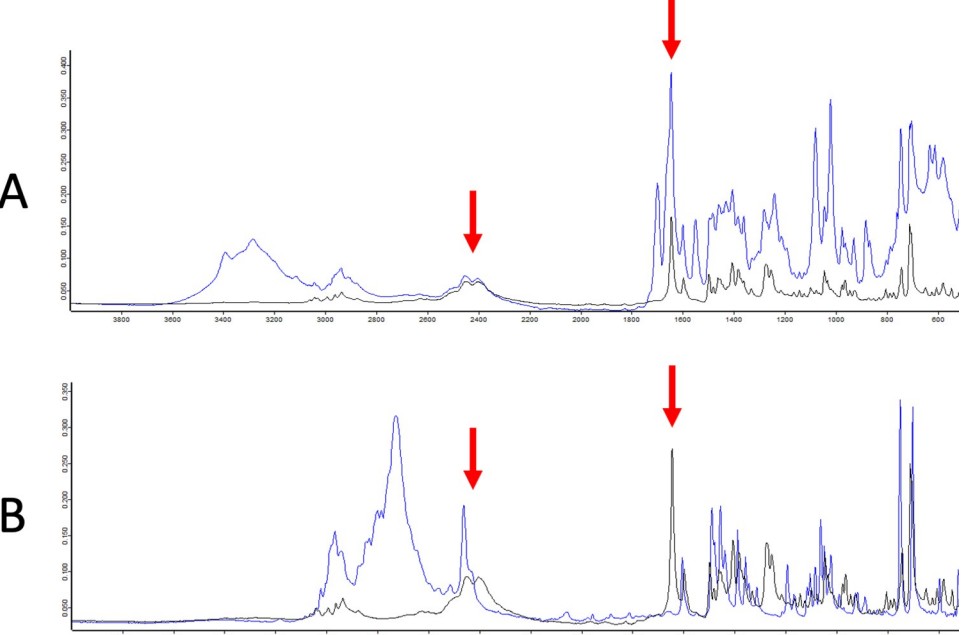

**Fig 1. Sample drug checking Fourier-transform infrared spectra.** Drug checking samples (blue) are overlaid with the fentanyl hydrochloride spectrum (black). The drug sample in A is an illicit opioid mixture containing fentanyl, caffeine, and mannitol. The drug sample in B is methamphetamine with no fentanyl present. The red arrows indicate characteristic peaks of fentanyl hydrochloride that are recognizable when present in drug mixtures.

In addition to the results obtained from the point-of-care technologies, the database also includes self-reported information about drug expectation and appearance (i.e., color, texture) [35]. Additionally, a separate and validated quantitative model using Bruker Quantitative Analysis 2 (QUANT 2) software was applied retrospectively to fentanyl-positive samples (identified via immunoassay strips) to generate approximate fentanyl concentrations for each sample [34]. The present study was restricted to samples obtained between August 2018 and January 2021, as linked data were available during this time period.

Participants provided verbal consent by agreeing to have their drugs checked and no personal identifying information was ever collected. This study was conducted as part of a larger drug checking evaluation project and was approved by the University of British Columbia/ Providence Health Care Research Ethics Board.

## Data preparation

In terms of input data, Joint Committee on Atomic and Molecular Physical Data exchange (jcamp-dx) encoded FTIR data files were converted to data tables containing wavenumbers and the corresponding absorbance for each spectrum (i.e., drug sample). Sample data tables meeting the following conditions were kept for analysis: first, the data table had at least 1,626 sampling points and second, the range of wavenumbers covered the range 600–3,800 cm$^{-1}$ including endpoints. A cubic spline was fitted to the samples by treating the absorbance as a function of the wavenumbers, which were then sampled from the cubic spline interpolated function at every second integer value of wavenumber starting from 600 to 3,798 cm$^{-1}$ which resulted in 1,600 features. Each sample was then standardized by subtracting the mean and dividing by the standard deviation of the discrete function.

Output data (i.e., reference standard) were recorded as the binary encoded results of the fentanyl immunoassay strips, which can consistently detect fentanyl and a range of analogues

[16, 17], including carfentanil, acetyl fentanyl, para-fluorofentanyl, and furanyl fentanyl [36]. We chose fentanyl test strips as the output data given the high accuracy for detecting fentanyl reported in previous validation studies and the lack of confirmatory analysis for a large proportion of the samples [15, 16]. Indeed, past research has shown that immunoassay strips can detect fentanyl with a sensitivity and specificity as high as 100% and 98%, respectively [16].

### Feature selection

Absorbance data were used as the sole feature for predicting the presence of fentanyl in samples. As part of the drug checking service, only minimal information is collected on drug samples to ensure that the service was low-barrier and wide-reaching. While additional sample-level data exist for the included samples, we chose to not include features such as colors and textures given that these are based on subjective human observable features and previous studies have shown that visual appearance of drug samples is not accurate in detecting the presence of fentanyl [37].

### Prediction methods and model selection

The goal of the study was to use anonymous, point-of-care drug checking data obtained from FTIR spectra to predict the presence of fentanyl and related analogues in drug samples. We implemented an artificial neural network-based model, which was composed of six layers in the following order (Fig 2):

1.  an input layer

2.  a one-dimensional convolution layer with eight filters, a kernel size of 12, and a rectified linear unit (ReLU) activation function

3.  a dropout layer with a 50% dropout rate

4.  a one-dimensional max pooling layer with a pool size of two

5.  a dense layer with output dimension 32 and a ReLU activation function

6.  a dense output layer with output dimension one and a sigmoid activation function.

The model was trained on 80% of the data and 20% of the data was reserved for testing the model's accuracy. At each epoch, the accuracy (i.e., number of correct predictions among total number of predictions) was measured on the test set and the model with the highest accuracy in the test set was kept.

We implemented the models and performed the study using the Python (version 3.9.6) programming language and the TensorFlow package (version 2.6.0).

### Statistical analyses

In assessing our model's performance, we used the prediction value which is a continuous variable ranging from 0 to 1. We applied a rounding procedure to transform this continuous output into a binary classification to align with our label values (i.e., the results from the corresponding fentanyl test strip for that sample). Specifically, prediction values greater than or equal to 0.5 were rounded up to 1, indicating a positive fentanyl result, while values less than 0.5 were rounded down to 0, indicating a negative result. These rounded prediction values were then compared against our label values, which represent the binary outcomes of the fentanyl test strip. Using these results, we calculated several diagnostic metrics, including F1

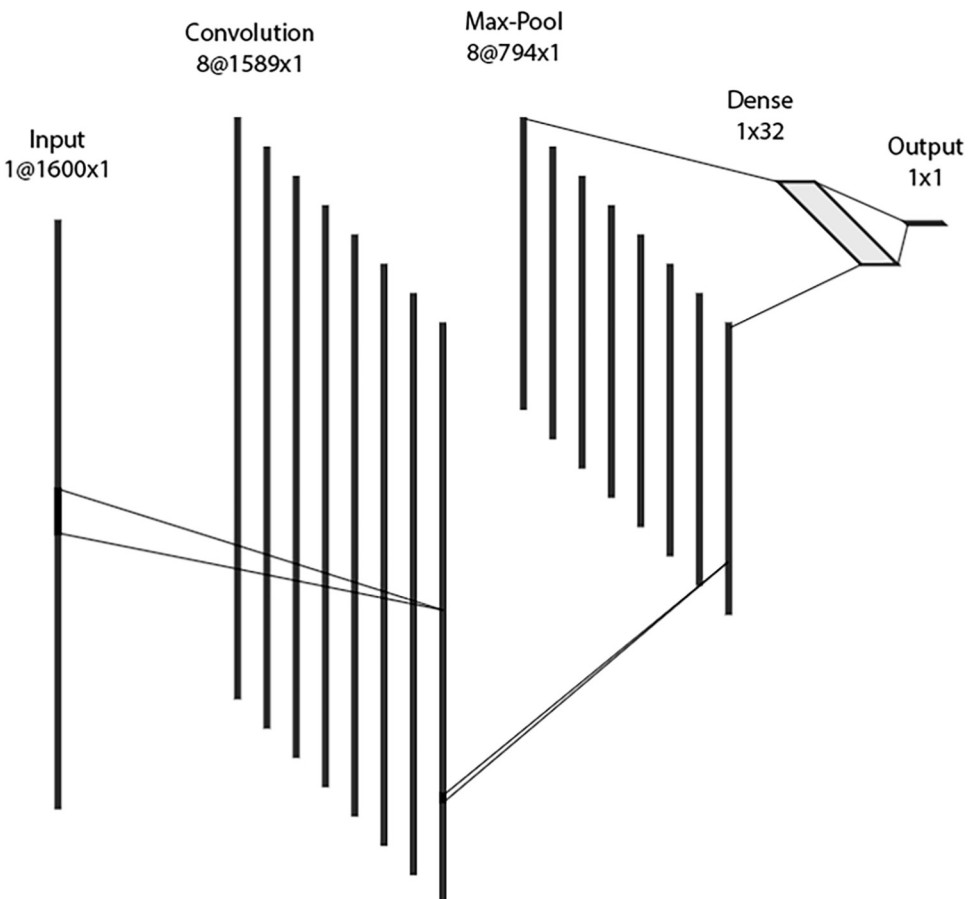

**Fig 2. A structural illustration of the final neural network model.** The network's input is 1600-dimensional and the remaining layers are 1589, 794, 32-dimensional with a 1-dimensional output.

score, accuracy, precision, recall, and area under the receiver-operating characteristic curve (AUROC). An F1 score describes a model's accuracy, taking the precision (correctly identified positives among test positives) and recall (correctly identified positives among true positives) into account. Given the imbalance in the dataset, calculating accuracy alone could be misleading; thus, the F1 score assessing the overall prediction performance using both precision and recall may be more appropriate. Bootstrapping methods based on 10,000 bootstrapped runs were used to create confidence intervals (CI) for the estimates. This methodology was chosen because of its ability to make strong inferences from the dataset, especially when the underlying distribution is unknown or not normal. As a secondary analysis, using the fentanyl concentration data available, we examined the descriptive statistics for fentanyl concentration among fentanyl positive samples that were identified by the neural network model but missed by the drug checking technician.

## Results

In total, 12,684 samples were included in our study: 6,099 (48.1%) were identified as fentanyl positive by immunoassay strips. Shown in Table 1, the large majority of samples were checked in the Vancouver Coastal Health Authority region (11,092; 87.4%). Among fentanyl immunoassay strip positive samples, the most common color and texture were purple (1,300; 23.0%)

**Table 1. Characteristics of the study sample, stratified by fentanyl immunoassay strip results (n = 12,684).**

| | Fentanyl immunoassay strip result | |
|---|---|---|
| | **Positive** | **Negative** |
| | **n = 6,099** | **n = 6,585** |
| **Year** | | |
| 2018 | 519 (8.5) | 394 (6.0) |
| 2019 | 2473 (40.5) | 1718 (26.1) |
| 2020 | 2723 (44.6) | 3960 (60.1) |
| 2021 | 384 (6.3) | 513 (7.8) |
| **Health Authority Region** | | |
| Fraser | 533 (8.7) | 194 (2.9) |
| Interior | 359 (5.9) | 445 (6.8) |
| Vancouver Coastal | 5161 (84.6) | 5931 (90.1) |
| Vancouver Island | 46 (0.8) | 15 (0.2) |
| **Color** | | |
| Black | 72 (1.2) | 28 (0.4) |
| Blue | 413 (6.8) | 138 (2.1) |
| Brown | 1173 (19.2) | 769 (11.7) |
| Colourless | 25 (0.4) | 1009 (15.3) |
| Green | 1229 (20.2) | 143 (2.2) |
| Grey | 292 (4.8) | 195 (3.0) |
| Orange | 217 (3.6) | 116 (1.8) |
| Pink | 368 (6.0) | 175 (2.7) |
| Purple | 1400 (23.0) | 167 (2.5) |
| Red | 140 (6.2) | 34 (0.5) |
| White | 380 (5.8) | 3552 (53.9) |
| Yellow | 351 (5.8) | 227 (3.4) |
| Other | 5 (0.1) | 0 (0.0) |
| Not reported | 34 (0.6) | 32 (0.5) |
| **Texture** | | |
| Chunk | 1471 (24.1) | 567 (8.6) |
| Crystal | 48 (0.8) | 2264 (34.4) |
| Flake | 18 (0.3) | 68 (1.0) |
| Granules | 533 (8.7) | 151 (2.3) |
| Liquid | 20 (0.3) | 110 (1.7) |
| Paste | 145 (2.4) | 49 (0.7) |
| Pebble | 3010 (49.4) | 271 (4.1) |
| Powder | 704 (11.5) | 2483 (37.7) |
| Pressed tablet | 7 (0.1) | 170 (2.6) |
| Residue | 65 (1.1) | 19 (0.3) |
| Tablet (pharmaceutical) | 65 (0.4) | 368 (5.6) |
| Other | 3 (0.05) | 28 (0.4) |
| Not reported | 23 (0.9) | 37 (0.6) |

**Note:** Categories where cells were less than 5 were recategorized into 'Other'

and pebbles (3,010; 40.4%), respectively, whereas for fentanyl immunoassay strip negative samples the most common color and texture were white (3,552; 53.9%) and powder (2,483; 37.7%), respectively.

**Table 2. Comparing the performance of different methods for detecting fentanyl.**

|  | F1 score | Accuracy | Precision | Recall |
|---|---|---|---|---|
| **Neural network model** | 96.4 | 96.4 | 95.7 | 97.1 |
| **% (95%CI)** | (95.6–97.1) | (95.6–97.1) | (94.5–96.7) | (96.1–98.0) |
| **Drug checking technician** | 78.4 | 82.4 | 99.4 | 64.7 |
| **% (95%CI)** | (76.4–80.3) | (80.9–83.9) | (98.8–99.9) | (62.1–67.3) |

CI: confidence interval

Using common diagnostic statistics to comprehensively evaluate the model's performance, we found that the neural network model outperformed results from those analyzed by a drug checking technician in metrics overall. Shown in Table 2, the neural network model achieved an F1 score of 96.4% (95% CI: 95.6–97.1) compared to 78.4% (95% CI: 76.4–80.3) by a technician. Regarding accuracy, the model achieved a score of 96.4% (95% CI: 95.6–97.1) compared to 82.4% (95% CI: 80.9–83.9) by a technician. For precision, the model achieved a score of 95.7% (95% CI: 94.5–96.7) compared to 99.4% (95% CI: 98. 8–99.9) by a technician. For recall, the model achieved a score of 97.1% (95% CI: 96.1–98.0) compared to 64.7% (95% CI: 62.1–67.3) by a technician. Lastly, as shown in Fig 3, the AUROC for the model was 99.0% (95% CI: 98.7–99.4).

In secondary analysis, there were a total of 2,117 samples that were correctly identified as fentanyl positive by the neural network model but were incorrectly labelled as fentanyl negative by the drug checking technician. Among those, the median fentanyl concentration was 2.3% (quartile 1–3: 0–4.6%).

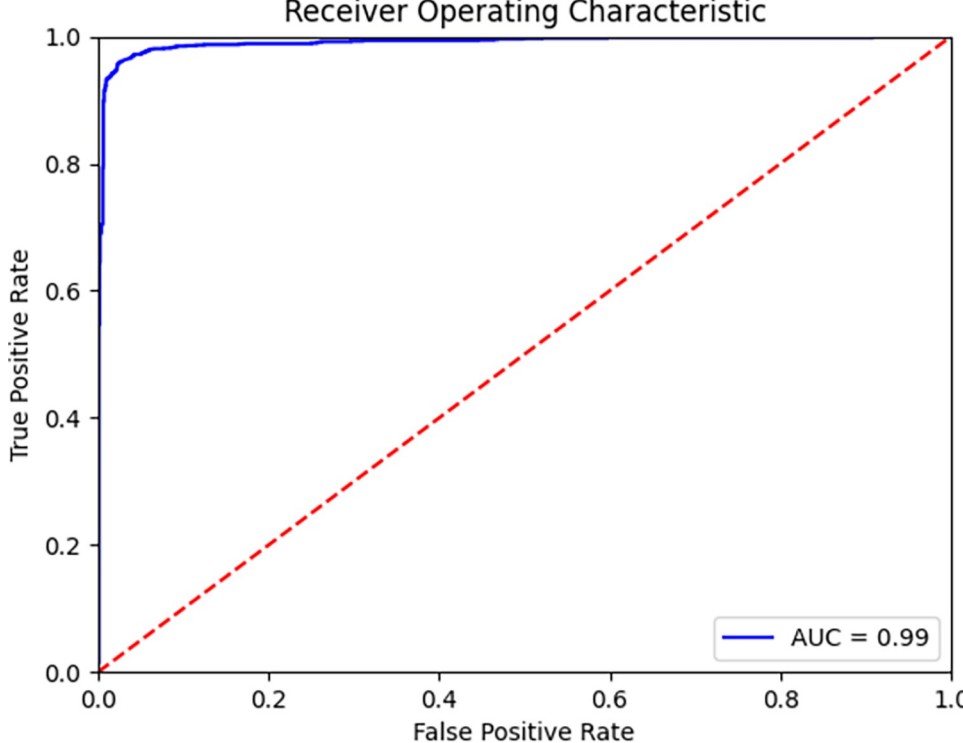

**Fig 3. Receiver-operating characteristic curve for the final neural network model for detecting fentanyl.**

## Discussion

In summary, our findings demonstrate that a neural network model was able to achieve highly promising and accurate results in predicting the presence of fentanyl in drug samples analyzed using FTIR data files, with an F1 score of 96.4%. Interestingly, the model was typically able to detect very low concentrations of fentanyl and analogues, with the majority of samples falling below the detection limit of FTIR spectroscopy [16, 17]. Previous data in this setting have indicated the emergence of highly potent fentanyl analogues (e.g., carfentanil) [3, 38, 39], which may be present in drug samples at very low concentrations which are not captured by trained technicians using FTIR. However, fentanyl immunoassay strips, with their high sensitivity and specificity to potent fentanyl analogues, account for this limitation of FTIR, hence the widespread use of these technologies in tandem [15, 16]. Thus, this study sheds light on the utility of a deep learning model to augment existing point-of-care drug checking technologies.

As indicated previously, many settings have adopted the use of combination FTIR spectroscopy and immunoassay strips as a drug checking intervention. Our findings show that given the high predictive performance of a neural network model for detecting fentanyl at low concentrations, it may be possible to replace fentanyl immunoassay strips with an automated model that can be used in conjunction with FTIR spectroscopy for services that already operate a spectrometer. Nevertheless, given that we only applied one feature (i.e., FTIR absorbance data) to the model to avoid subjective noise, future research should seek to explore additional characteristics of drug checking samples (e.g., appearance) that can be used to optimize the model's prediction performance. It may also be worthwhile to retrain the model on an ongoing basis as novel psychoactive substances, including novel synthetic opioids, are identified in the highly dynamic, unregulated drug supply. Future research should also seek to examine whether a machine learning model can accurately quantify fentanyl concentration in samples, as this information may be more valuable in a fentanyl-saturated unregulated drug market.

There are some limitations to consider. The neural network model was applied to drug samples checked in BC; as such, the model may not be generalizable to other settings where the unregulated drug supply may be vastly different. For example, in settings where different cutting agents are used, equivalent amounts of fentanyl appear different with FTIR analysis. Future research with additional sources of data, such as those from outside of BC, will determine if or how our results may differ from the unregulated drug supply internationally. Given that the point-of-care technologies were limited in their ability to differentiate between fentanyl and its analogues at low concentrations, we were unable to confirm whether the model captured fentanyl analogues or if the model only captured fentanyl. Lastly, recent studies have suggested that depending on the composition and concentration of drug samples, line faintness on immunoassay strips (which were used as the reference standard for our model) may lead to subjectivity in determining the presence of fentanyl [40, 41]. Nevertheless, our study employed trained drug checking technicians to interpret the immunoassay strips and other studies have shown the immunoassay strips can perform with high sensitivity and specificity [15, 16].

## Conclusion

In sum, we developed a neural network model that can accurately predict the presence of fentanyl using absorbance spectrum data files obtained from FTIR spectroscopy. The model was able to correctly detect fentanyl-positive samples below the limit of detection of the FTIR spectrometer. Our findings point to the potential of integrating this tool within drug checking services utilizing FTIR spectroscopy to improve decision making and reduce harms associated with overdose and other negative health outcomes.

## Acknowledgments

We offer thanks to those individuals who participated directly in the study by having their drugs analyzed, with the hopes that this involvement will contribute to utilizable public health information, improved harm reduction care, and potentially, decreased loss of life. We would also like to thank researchers and staff at various community organizations, health authorities, and laboratory services across the province for their work in this area. Health Canada Drug Analysis Service provided confirmatory testing services; however, the findings reported here should in no way be taken as an endorsement of the specific point-of-care technologies that were used for this study.

## Author Contributions

**Conceptualization:** Lianping Ti, Brandon D. L. Marshall.

**Data curation:** Cameron J. Grant, Samuel Tobias.

**Formal analysis:** Lianping Ti, Cameron J. Grant.

**Funding acquisition:** Lianping Ti.

**Project administration:** Lianping Ti, Samuel Tobias.

**Resources:** Richard Laing.

**Supervision:** Lianping Ti.

**Writing – original draft:** Lianping Ti, Brandon D. L. Marshall.

**Writing – review & editing:** Lianping Ti, Cameron J. Grant, Samuel Tobias, Dennis K. Hore, Richard Laing.

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
