## [Decision Letter · Decision Letter 0]

11 Apr 2023

PONE-D-23-02432Development of a neural network model to predict the presence of fentanyl in community drug samplesPLOS ONE

Dear Dr. Ti,

Thank you for submitting your manuscript to PLOS ONE. After careful consideration, we feel that it has merit but does not fully meet PLOS ONE’s publication criteria as it currently stands. Therefore, we invite you to submit a revised version of the manuscript that addresses the points raised during the review process. Please submit your revised manuscript by May 26 2023 11:59PM. If you will need more time than this to complete your revisions, please reply to this message or contact the journal office at plosone@plos.org. Please include the following items when submitting your revised manuscript:A rebuttal letter that responds to each point raised by the academic editor and reviewer(s). You should upload this letter as a separate file labeled 'Response to Reviewers'.A marked-up copy of your manuscript that highlights changes made to the original version. You should upload this as a separate file labeled 'Revised Manuscript with Track Changes'.An unmarked version of your revised paper without tracked changes. You should upload this as a separate file labeled 'Manuscript'.

We look forward to receiving your revised manuscript.

Kind regards,

Muhammad Hanif

Academic Editor

PLOS ONE

Journal Requirements:

2. Please ensure that you have specified (1) whether consent was informed and (2) what type you obtained (for instance, written or verbal, and if verbal, how it was documented and witnessed). If your study included minors, state whether you obtained consent from parents or guardians. If the need for consent was waived by the ethics committee, please include this information.

"The study was supported by the Health Canada Substance Use and Addictions Program (1718-HQ-000024; https://www.canada.ca/en/health-canada/services/substance-use/canadian-drugs-substances-strategy/funding/substance-use-addictions-program.html), Vancouver Foundation (https://www.vancouverfoundation.ca), and the US National Institutes of Health-National Institute on Drug Abuse (R01DA052381; https://nida.nih.gov). The content is solely the responsibility of the authors and does not necessarily represent the official views of these funding agencies. LT is supported by a Michael Smith Health Research British Columbia (MSHRBC; https://healthresearchbc.ca) Scholar Award."

We note that one or more of the authors is affiliated with the funding organization, indicating the funder may have had some role in the design, data collection, analysis or preparation of your manuscript for publication; in other words, the funder played an indirect role through the participation of the co-authors. If the funding organization did not play a role in the study design, data collection and analysis, decision to publish, or preparation of the manuscript and only provided financial support in the form of authors' salaries and/or research materials, please do the following:

(1) Review your statements relating to the author contributions, and ensure you have specifically and accurately indicated the role(s) that these authors had in your study. These amendments should be made in the online form.

(2) Confirm in your cover letter that you agree with the following statement, and we will change the online submission form on your behalf: 

5. Thank you for stating the following in your Competing Interests section:  "No authors have competing interests."

6. We note that you have indicated that data from this study are available upon request. PLOS only allows data to be available upon request if there are legal or ethical restrictions on sharing data publicly. For more information on unacceptable data access restrictions, please see http://journals.plos.org/plosone/s/data-availability#loc-unacceptable-data-access-restrictions. 

Reviewers' comments:

Reviewer's Responses to Questions

**Comments to the Author**

1. Is the manuscript technically sound, and do the data support the conclusions?

Reviewer #1: Partly

Reviewer #2: Yes

Reviewer #3: Yes

Reviewer #4: No

2. Has the statistical analysis been performed appropriately and rigorously? 

Reviewer #1: No

Reviewer #2: Yes

Reviewer #3: No

Reviewer #4: No

3. Have the authors made all data underlying the findings in their manuscript fully available?

Reviewer #1: Yes

Reviewer #2: No

Reviewer #3: Yes

Reviewer #4: No

4. Is the manuscript presented in an intelligible fashion and written in standard English?

Reviewer #1: No

Reviewer #2: Yes

Reviewer #3: Yes

Reviewer #4: No

5. Review Comments to the Author

Reviewer #1: 1.English grammar needs corrections

2.Statistical analysis needs more details

3.analyzed using FTIR spectra files ???

4.FTIR peaks which

4. Missing the main peaks indicated this space

5.References needs to be coorected for style specific

Reviewer #2: The manuscript is a technically sound piece of scientific research with data that supports the conclusions. Experiments were performed with appropriate sample sizes. The objective of this study was to develop a neural network model to identify fentanyl and related analogues more accurately in drug samples compared to traditional analysis by technicians. They discussed that Neural network models can accurately predict the presence of fentanyl and related analogues using FTIR data, including samples with low fentanyl concentrations. Integrating this tool within drug checking services utilizing FTIR spectroscopy has the potential to improve decision making to reduce the risk of overdose and other negative health outcomes and concluded that their findings point to the potential of integrating machine learning within drug checking services utilizing FTIR spectroscopy to improve decision making and reduce harms associated with overdose and other negative health outcomes.

FTIR is being used in developed economies for quantitative analysis and its susrquent integration with machine learning methods can be helpful

Though neural network modelling is a san-statistical approach, but authors have applied required stat while comparing supplementary groups.

Manuscript was presented in a precised and concised way with a good standard of english.

Reviewer #3: Manuscript Number: PONE-D-23-02432, Reviewers Comments:

The paper presented in this study is “Development of a neural network model to predict the presence of fentanyl in community drug samples”. The paper is clearly written and well organized. The introduction and background are reasonable given the premise of the paper. Figures and tables are comprehensive and helpful. The problem statement and objectives are clearly defined and explained with the help of different instrumentations. At the end of the introduction; it would be helpful to add more information of your current study and clear statements of the objective and results of this study. Such a statement would provide a transition to the main ideas being presented. Conclusion section could be improved to better reflect the large amount of information reviewed in relation to the title/objective of the paper. In my view, the conclusions should be expanded to better summarize the overall "feel" of the main review section to give the reader a strong message. Some of the references are outdated. However, before I can recommend its publication, the authors should address the following questions

Some questions to author

1. Abstract was too long of 325 words try to summarize to maximum 250 words

2. Keywords seemed to be quite unimportant like “harm reduction” etc and few so should be revised and add more to number of six.

3. The term drug checking in the manuscript is not appropriate. Is this term was from any scientific source please add reference.

4. Line 61 to 71 sentence is too long and did not understand what author wanted to say. Please clarify and rewrite such kind of sentence.

5. Line 122 (a significant number of lives had already been claimed) is not very clear.

6. Line 143-145 the author mentioned fentanyl positive samples but did not clear how these samples were obtained and what kind of samples it was.

7. The author described in manuscript FTIR analysis but it was quite difficult to understand that FTIR only describes presence of specific peak of compound but it does not give information about quantity of compound.

8. The manuscript needs to include more description about artificial neural network.

9. Figure 2 does not explain any information about characteristics

10. Conclusion should be separate from discussion

11. No graphical representation of full factorial design was seen in the manuscript.

General Assessment

1. The method presented here is not thrillingly novel and distinctly superior.

2. The material used here is not excitingly novel.

3. The article does not have wider scope and applicability for reader of PLOS ONE.

4. The method is not the selective enough over chemically related drugs.

Reviewer #4: Following are my comments for the article titled “Development of a neural network model to predict the presence of fentanyl in community drug samples”

1. There are other similar research work published in this area and for this compound like(Chen et al., 2022; Xu et al., 2020) therefore authors should discuss whats new compared to other similar work

Chen, H., Kim, S., Hardie, J.M., Thirumalaraju, P., Gharpure, S., Rostamian, S., Udayakumar, S., Lei, Q., Cho, G., Kanakasabapathy, M.K., Shafiee, H., 2022. Deep learning-assisted sensitive detection of fentanyl using a bubbling-microchip. Lab on a Chip 22, 4531-4540.

Xu, M., Wang, C.-H., Terracciano, A.C., Masunov, A.E., Vasu, S.S., 2020. High accuracy machine learning identification of fentanyl-relevant molecular compound classification via constituent functional group analysis. Scientific Reports 10, 13569.

2. There are more sensitive techniques like LC-MS/MS and GC-MS and considered as the gold standard techniques for estimating fentanyl in blood samples due to their high sensitivity and specificity. FTIR (Fourier Transform Infrared Spectroscopy) is not typically used in diagnostic labs for fentanyl toxicity testing. FTIR is a powerful technique that is commonly used in materials science, chemistry, and forensic analysis, but it is not well-suited for the detection and quantification of small molecules like fentanyl in biological samples. The techniques such as LC-MS/MS, GC-MS, ELISA, HPLC, and CE, are more commonly used for detecting and quantifying fentanyl in biological samples such as blood, urine, and hair.

3. Manuscript is poorly written, though the information is given that the model is developed but it not properly explained. More Tables and Figures should be added for the better understanding of the model. When writing a Research article it should be written in a way that it can be replicated.

4. There is no description about the drug structure, FTIR spectra etc .

6. PLOS authors have the option to publish the peer review history of their article (what does this mean?). If published, this will include your full peer review and any attached files.

Reviewer #1: No

Reviewer #2: **Yes: **Dr. Asad Majeed Khan

Reviewer #3: No

Reviewer #4: No

---

## [Author Response · Author response to Decision Letter 0]

1 Jun 2023

Response to Reviewer 1:

We thank Reviewer 1 for their succinct comments about our manuscript. We have responded to Reviewer 1’s individual comments using numbered bullets below.

1. We thank Reviewer 1 for the comment regarding the need for grammatical corrections. The updated version of the manuscript has received thorough proofreading and we feel it is free of grammatical errors. 

2. We thank Reviewer 1 for their suggestion that the “statistical analysis needs more details.” In light of this comment, we have included the following information that further describes the statistical analysis portion of methods: 

In assessing our model's performance, we utilized the prediction value which is a continuous variable ranging from 0 to 1. We applied a rounding procedure to transform this continuous output into a binary classification to align with our label values (the results from the corresponding fentanyl test strip for that sample). Specifically, prediction values greater than or equal to 0.5 were rounded up to 1, indicating a positive fentanyl result, while values less than 0.5 were rounded down to 0, indicating a negative result. These rounded prediction values were then compared against our label values, which represent the binary outcomes of the fentanyl test strip. Using these results, we calculated several diagnostic metrics, including F1 score, accuracy, precision, recall, and area under the receiver-operating characteristic curve (AUROC). An F1 score describes a model’s accuracy, taking the precision (correctly identified positives among test positives) and recall (correctly identified positives among true positives) into account.

3. We thank Reviewer 1 for raising the issue of ambiguity regarding how we speak of file types. We have removed reference to “FTIR spectra files” and replaced it with the following:

…(jcamp-dx) encoded FTIR data files were converted to data tables…

4. We apologize to Reviewer 1 and the Editor that we are unable to determine what Reviewer 1 meant by the comment, “4. FTIR peaks which 4. Missing the main peaks indicated this space.” We hope that with the revisions in the updated manuscript, this comment has been sufficiently addressed.

5. We thank Reviewer 1 for highlighting the need to ensure consistency in reference style. As a part of our proofreading process, we have ensured consistency in reference formatting.

Response to Reviewer 2:

We thank Reviewer 2 for their time and generous comments regarding our manuscript. Reviewer 2 commented that “the manuscript is a technically sound piece of scientific research” and raised no concerns or need for any edits.

Response to Reviewer 3:

We thank Reviewer 3 for their comments about our manuscript. We have responded to Reviewer 3’s numbered comments below:

1. We appreciate Reviewer 3’s comments about the abstract being too long. While the abstract length is within the limit outlined in the PLOS ONE submission guidelines, we are happy to shorten it if the Editor deems it necessary.

2. We thank Reviewer 3 for noting additional keywords are beneficial. We do, however, feel that the keyword “harm reduction” is relevant to this article. We have added the additional keywords of “deep learning” and “FTIR spectroscopy.”

3. We thank Reviewer 3 for their comment regarding the use of the term “drug checking” in the manuscript. Drug checking is the internationally recognized term to refer to the harm reduction intervention of analyzing illicit drugs to determine what they contain before consumption. We feel the use of this specific term is adequately defined with supporting references in the manuscript: 

In response, drug checking services have been implemented as an important harm reduction intervention for people who use drugs to chemically analyze their substances and receive fact-based information and consultation regarding the compounds detected in their sample [4–6]. Not a new concept, the vast majority of drug checking services have previously focused on drugs used in party and festival settings (e.g., psychedelics, stimulants) [7,8], and less is known about drug checking in the context of the synthetic opioid-driven overdose epidemic. While limited, recent studies have started to show a positive impact of drug checking, including increased engagement in overdose risk reduction practices following the use of the service in some communities [5,9–11].

4. Unfortunately, we are unable to discern which specific sentence Reviewer 3 is referencing in their 4th comment, as the line numbers referenced do not contain sentences, but instead are the manuscript keywords. We apologize to Reviewer 3 and hope that during the revisions stage their concern has been adequately addressed otherwise. 

5. We thank Reviewer 3 for their comment regarding the clarity of the use of the term “claimed” to refer to loss of life. We have amended the sentence to the following:

By the time a public health emergency was declared in 2016, a significant number of lives had already been lost to illicit drug toxicity, largely due to the widespread presence of fentanyl and other novel synthetic opioids in the unregulated drug supply [29].

6. We appreciate Reviewer 3’s comment about the clarification needed on how fentanyl-positive samples were obtained. The manuscript has been updated to provide further detail: 

Further details about the drug checking service and the use of these technologies, which have been operating since October 2017, have been described elsewhere in detail [15,20]. Briefly, individuals in harm reduction sites anonymously provided an approximately 5 mg drug sample to a trained drug checking technician and information about what they believed the sample to contain [34]. The drug sample was then analyzed with FTIR and immunoassay strips, with results being immediately available to the person who provided the sample. Results from FTIR analysis include compounds detected in abundant amounts (typically those 5% by weight and above), including both active ingredients (e.g., fentanyl, heroin, methamphetamine) and inert cutting agents (e.g., mannitol, inositol, lactose) [6,17,20].

7. We thank Reviewer 3 for their comment about details of FTIR analysis and how it does not give information about quantity of specific compounds. To address this, we have added information to the methods section about the results generated by FTIR analysis at point of care:

The drug sample was then analyzed with FTIR and immunoassay strips, with results being immediately available to the person who provided the sample. Results from FTIR analysis include compounds detected in abundant amounts (typically those 5% by weight and above), including both active ingredients (e.g., fentanyl, heroin, methamphetamine) and inert cutting agents (e.g., mannitol, inositol, lactose) [6,17,20].

8. We thank Reviewer 3 for the comment about how the manuscript would benefit from further description about the artificial neural network. To address this comment, we have restructured and added additional text to the “Prediction methods and model selection” subsection of the methods section:

The goal of the study was to use anonymous, point-of-care drug checking data obtained from FTIR spectra to predict the presence of fentanyl and related analogues in drug samples. We implemented an artificial neural network-based model, which was composed of six layers in the following order (Fig 1):

1) an input layer

2) a one-dimensional convolution layer with eight filters, a kernel size of 12, and a rectified linear unit (ReLU) activation function

3) a dropout layer with a 50% dropout rate

4) a one-dimensional max pooling layer with a pool size of two

5) a dense layer with output dimension 32 and a ReLU activation function

6) a dense output layer with output dimension one and a sigmoid activation function. 

The model was trained on 80% of the data and 20% of the data was reserved for testing the model's accuracy. At each epoch, the accuracy (i.e., number of correct predictions among total number of predictions) was measured on the test set, and the model with the highest accuracy in the test set was kept. 

9. We thank Reviewer 3 for their suggestion about how to improve Fig 2 (now Fig 3). We feel that the ROC curve presented in Fig 2 sufficiently describes the performance classification characteristics that it seeks to depict. If the Editor feels that further description of characteristics is warranted, we can address this comment in another fashion. 

10. We thank Reviewer 3 for suggesting how the conclusions should be separated from the discussion section. We have separated the two sections in the manuscript.

11. We thank Reviewer 3 for their comment regarding a graphical representation of a full factorial design. We have included a graphical representation of the model (Fig 2) and we hope that this graphic helps illustrate the final model developed over the course of the present study. 

Response to Reviewer 4:

1. We thank Reviewer 4 for identifying previous research involving machine learning to identify fentanyl in various forms. Unfortunately, these articles are not related to the identification of fentanyl in illicit drug samples, but instead one is related to biological fluids and the other is related to chemical structure analysis. We feel these previous articles do not detract from the novel findings of our research, as it remains the first in published literature to describe the use of deep learning to detect fentanyl in illicit drug samples directly. 

2. We thank Reviewer 4 for sharing their expertise regarding the use of different (and more sensitive) technologies to detect fentanyl in biologic samples. We have included new text that describes the advantages and disadvantages of using such technologies in a point-of-care drug checking setting to address the unregulated drug supply. We have also provided further information on the FTIR. 

Drug checking can be conducted using a range of technologies, offering both benefits and challenges in terms of accuracy, timing, cost, portability, ability to detect a range of compounds, preparation, and required human resources [4,6,12,13]. For example, gas chromatography/mass spectrometry (GC-MS) is highly sensitive, but is expensive, time-consuming, and requires a trained laboratory technician to operate the machine [6,14]. In contrast, immunoassay strips are inexpensive, easy to use, portable, and can provide results in less than five minutes, but they are unable to detect a range of compounds and can only produce qualitative results (i.e., presence, absence) [15,16]. Fourier-transform infrared (FTIR) spectroscopy is being implemented across North America for drug checking given the ease and speed of sample preparation and analysis, portability, and ability to detect a range of compounds [15,16]. However, recent validation studies have indicated that FTIR spectroscopy can only detect compounds above a certain limit; for fentanyl and some fentanyl analogues, that limit is 3-10% quantity by weight, depending on the setting and the presence of specific cuts and buffs [16,17]. Given that some analogues (e.g., carfentanil) can cause toxicity at very low concentrations [18,19], failing to detect these compounds when present is a substantial concern when using FTIR spectroscopy on its own.

…

Further details about the drug checking service and the use of these technologies, which have been operating since October 2017, have been described elsewhere in detail [15,20]. Briefly, individuals in harm reduction sites anonymously provided an approximately 5 mg drug sample to a trained drug checking technician and information about what they believed the sample to contain [34]. The drug sample was then analyzed with FTIR and immunoassay strips, with results being immediately available to the person who provided the sample. Results from FTIR analysis include compounds detected in abundant amounts (typically those 5% by weight and above), including both active ingredients (e.g., fentanyl, heroin, methamphetamine) and inert cutting agents (e.g., mannitol, inositol, lactose) [6,17,20].

As well, we have described in more detail the use of FTIR spectroscopy more generally, in addition to including a new figure that depicts a typical FTIR spectrum (per Reviewer 4’s fourth comment).

3. We thank Reviewer 4 for their comment about providing further details on the methods to allow for replicability of the manuscript. We feel the manuscript, particularly with the new amendments in the course of the revisions process, have provided sufficient instruction to PLOS ONE readers should they wish to replicate the study, including new text and restructuring the “Prediction methods and model selection”. Upon further review, should the Editor feel further enhancements are necessary, we defer to their decision. 

4. We agree with Reviewer 4’s comment that the manuscript lacks description of the structure of the fentanyl molecule and what an FTIR spectrum is and looks like. To address this issue, we have added an additional figure to the manuscript that depicts what a typical FTIR spectrum of a sample in our analysis looks like. The following text has also been added to the manuscript to more thoroughly describe FTIR spectra. 

An illustrative example of what a drug checking technician analyzes (and attempts to determine if fentanyl is present) is displayed in Fig 1.

[Figure caption] Fig 1. Sample drug checking Fourier-transform infrared spectra. Drug checking samples (blue) are overlaid with the fentanyl hydrochloride spectrum (black). The drug sample in A is an illicit opioid mixture containing fentanyl, caffeine, and mannitol. The drug sample in B is methamphetamine with no fentanyl present. The red arrows indicate characteristic peaks of fentanyl hydrochloride that are recognizable when present in drug mixtures.

---

## [Editor Report · Decision Letter 1]

3 Jul 2023

Development of a neural network model to predict the presence of fentanyl in community drug samples

PONE-D-23-02432R1

Dear Dr. Ti,

We’re pleased to inform you that your manuscript has been judged scientifically suitable for publication and will be formally accepted for publication once it meets all outstanding technical requirements.

Kind regards,

Muhammad Hanif

Academic Editor

PLOS ONE

Additional Editor Comments (optional):

I have gone trough the revision submitted by the authors and found that all the points have been addressed
---

## [Editor Report · Acceptance letter]

5 Jul 2023

PONE-D-23-02432R1 

Development of a neural network model to predict the presence of fentanyl in community drug samples 

Dear Dr. Ti:

I'm pleased to inform you that your manuscript has been deemed suitable for publication in PLOS ONE. Congratulations! Your manuscript is now with our production department. 

Kind regards, 

on behalf of

Dr. Muhammad Hanif 

Academic Editor

PLOS ONE